# Are Rurality, Area Deprivation, Access to Outside Space, and Green Space Associated with Mental Health during the COVID-19 Pandemic? A Cross Sectional Study (CHARIS-E)

**DOI:** 10.3390/ijerph18083869

**Published:** 2021-04-07

**Authors:** Gill Hubbard, Chantal den Daas, Marie Johnston, Peter Murchie, Catharine Ward Thompson, Diane Dixon

**Affiliations:** 1Centre for Health Sciences, Department of Nursing and Midwifery, University of the Highlands and Islands, Inverness 999020, UK; 2Health Psychology Group, Institute of Applied Health Sciences, University of Aberdeen, Aberdeen 999020, UK; chantal.dendaas@abdn.ac.uk (C.d.D.); m.johnston@abdn.ac.uk (M.J.); diane.dixon@abdn.ac.uk (D.D.); 3Centre of Academic Primary Care, Institute of Applied Health Sciences, University of Aberdeen, Aberdeen AB24 3FX, UK; p.murchie@abdn.ac.uk; 4OPENSpace Research Centre, University of Edinburgh, Edinburgh EH3 9DF, UK; c.ward-thompson@ed.ac.uk

**Keywords:** COVID-19, environment, place, rural, urban, area deprivation, green space, mental health, pandemic

## Abstract

The study investigated if rurality, area deprivation, access to outside space (Study 1), and frequency of visiting and duration in green space (Study 2) are associated with mental health during the COVID-19 pandemic and examined if individual demographics (age, gender, COVID-19 shielding status) and illness beliefs have a direct association with mental health during the COVID-19 pandemic. A serial, weekly, nationally representative, cross-sectional, observational study of randomly selected adults was conducted in Scotland during June and July 2020. If available, validated instruments were used to measure psychological distress, individual demographics, illness beliefs, and the following characteristics: Rurality, area deprivation, access to residential outside space, frequency of visiting, and duration in green space. Simple linear regressions followed by examination of moderation effect were conducted. There were 2969 participants in Study 1, of which 1765 (59.6%) were female, 349 (11.9%) were in the shielding category, and the median age was 54 years. There were 502 participants in Study 2, of which 295 (58.60%) were female, 58 (11.6%) were in shielding category, and the median age was 53 years. Direct effects showed that psychological distress was worse if participants reported the following: Urban, in a deprived area, no access to or sharing residential outside space, fewer visits to green space (environment), younger, female, in the shielding category (demographics), worse illness (COVID-19) representations, and greater threat perception (illness beliefs). Moderation analyses showed that environmental factors amplified the direct effects of the individual factors on psychological distress. This study offers pointers for public health and for environmental planning, design, and management, including housing design and public open space provision and regulation.

## 1. Introduction

During the COVID-19 pandemic, governments have introduced legislation and guidelines to prevent the spread of the coronavirus, which have restricted movement and social interactions and have frequently confined people within small geographical areas. Prior to the pandemic, various indices of the environment have been associated with mental health [1,2]. Hence, one might expect relationships between the environment and mental health to be particularly salient during the current pandemic. Environment is a concept with several definitions and dimensions [3], including rurality, area deprivation, residential outside space, and green and natural environments. However, there are other factors related to mental health that may also be important in the context of the pandemic.

There is ample evidence that individual demographic factors are related to mental health. During the COVID-19 pandemic, for instance, women have consistently reported more anxiety and depression than men and young people have reported greater anxiety than older people [4,5,6]. In addition, illness beliefs have been associated with mental health during the current pandemic. A growing number of studies have used the concept of risk perception to understand the effects of the pandemic on mental health [6,7,8,9,10]. These studies have shown that different dimensions of the construct risk perception are associated with mental health. Other beliefs, such as beliefs about the illness COVID-19, have also been found to be associated with mental health during the pandemic [6], which is consistent with the substantial body of empirical work based on the Common Sense Self-Regulation Model (CS-SRM) [11] and with the CS-SRM as theoretically conceived [12]. CS-SRM proposes that people construct representations of a health threat, which help them make sense of their experiences and provides a basis for their own coping responses. Beliefs about illness are central to the model and incorporate five key components: Beliefs about the nature (identity), time-course (timeline), personal impact (consequences), causal factors (cause), and feasibility of control or cure (control/cure) of the illness

It is conceivable that different environments would amplify the effects of these individual demographic factors and illness beliefs on mental health during the COVID-19 pandemic. Systematic reviews have concluded that characteristics of the environment are associated with depressive mood and psychological distress [13,14]. Associations have been found between area deprivation and death by suicide, anxiety, and depression [15,16,17], but other studies have suggested that this can be explained by individual (e.g., age, gender, marital status) rather than environmental factors [17,18]. Systematic reviews and meta-analyses have highlighted differences between people living in rural and urban areas for depression and psychiatric disorders, with some studies showing that mental health is worse in urban compared to rural areas and other studies suggesting the opposite [19,20]. The availability of green space might be one important factor in explaining urban-rural differences in mental health [21] and the link between area deprivation and stress [22,23]. Several theoretical hypotheses that have been proposed to explain how exposure to the natural environment may impact mental wellbeing [24,25]. These include a suggestion that humans have an innate evolutionary connection to the natural environment such that any exposure to it will reduce levels of stress and improve mood (“biophilia hypothesis”) [26,27] and a conceptualization of natural places as ‘restorative environments’ that mediate the negative effects of stress (“psycho-evolutionary stress reduction theory”) [28].

A pre-pandemic study found that people spending ≥120 min per week in natural environments reported good health and well-being compared to people with no contact with nature and that positive associations peaked at between 200–300 min per week with no further gain [29]. The extent to which people use green space during the pandemic may have changed due to the temporary closure of parks and golf courses and restrictions on travel and opportunities to meet with others. Nonetheless, uncertainties about the influence of environmental factors on mental health remain [30]. For example, a systematic review of urban green space exposure reported mixed results with associations found between green space and mood but not for depression and stress [31].

Taken together, the above body of work calls for the examination of the impact of environment both directly and in moderating individual demographic and psychological factors that are associated with mental health during pandemics. The research may be useful to health care planners by highlighting which modifiable environmental factors could be used to improve mental health during pandemics. To date, there are few studies about the relationship between environment and mental health in the context of pandemics, yet home lockdown and restrictions on travel during the COVID-19 pandemic have heightened the importance of the environment as an aspect of people’s lives. This explains the focus of our paper.

An aim of the COVID-19 Health and Adherence Research in Scotland (CHARIS) project [32] was to investigate mental health in the adult population during the COVID-19 pandemic and to explain variations in mental health. Hence, two CHARIS studies examined the influence of environment on mental health, which are collectively referred to as CHARIS-E, with E signifying environment.

The aims of Study 1 and 2 were to: (a) Confirm if individual demographics (age, gender, COVID-19 shielding status (people with specific medical conditions were identified by clinicians as at high risk of severe illness from COVID-19 and were placed on a national ‘shielding list’ and advised to take extra precautions to reduce their risk of infection)) and illness beliefs have a direct association with mental health during the COVID-19 pandemic; (b) determine if the following environmental factors—rurality, area deprivation, access to residential outside space (Study 1), and frequency and duration of visiting green space (Study 2)—have a direct association with mental health during the COVID-19 pandemic; and (c) determine if any of these environmental factors moderate the relationship between individual demographics and illness beliefs and mental health during the COVID-19 pandemic.

## 2. Materials and Methods

### 2.1. Design

The CHARIS project was a serial, weekly, nationally representative, cross-sectional, observational study of randomly selected adults in Scotland. CHARIS-E Study 1 draws on data collected for 6 consecutive weeks in June and July 2020, while CHARIS-E study 2 draws on data collected for one week at the beginning of July 2020. These studies were conducted at the point when Scotland had just come out of lockdown [33]. Key contextual changes relevant to this study include the 5-mile restriction on travel distance being lifted (3rd July).

Ethical approval for this study was granted by the Life Sciences and Medicine College Ethics Review Board (CERB) at the University where the Principal Investigator for CHARIS was employed.

### 2.2. Setting

Scotland has a total population of 5.4 million, 83% of whom are adults. Rural Scotland accounts for 98% of the land mass of Scotland and nearly one-fifth of the population are residents there [34]. Of the population, 70% live in the predominantly urban Central Belt of Scotland. The most deprived areas are in urban areas and the West of Scotland, for example, Glasgow city has 44.4% of the most deprived areas and Aberdeenshire the least, with 2.6% [35]. Whereas c. 65% of Scottish adults live within a 5-minute walk of their nearest green or blue space [36], a smaller proportion of adults in deprived areas live within a 5-minute walk compared to adults in the least deprived areas (58% cf. 68% [37]).

### 2.3. Participants

All adult men and women aged 16 or older, able to speak English, and currently living in Scotland were eligible to participate. No other exclusion criteria were applied. The CHARIS project was administered by a commercial polling company (Ipsos MORI Scotland) who sampled participants using random digit dialing to landlines and targeted mobiles. Each week, 500 participants were sampled. Quotas were applied to ensure that a representative sample of Scotland adults was achieved. Quotas were based on gender (52% female), age, working status (42% working fulltime), and geographical locations (distribution over the Scottish Parliament regions). A leeway on the quotas (approximately 30%) was allowed to help ensure the overall sample was achieved in a reasonable time.

### 2.4. Variables and Measures

#### 2.4.1. Psychological Distress (Study 1 and 2)

Psychological distress was measured using the 4-item Patient Health Questionnaire (PHQ-4), which is an ultra-brief screening scale for anxiety and depression [38]. The introductory text was slightly adapted for a telephone as opposed to a written administered survey. Participants were asked: “Over the last 2 weeks, how often have you been bothered by the following problems? Tell me which answer option best applies: (1) Feeling nervous, anxious, or on edge, (2) Not being able to stop or control worrying, (3) Feeling down, depressed, or hopeless, (4) Little interest or pleasure in doing things.” For each item, participants were given the following response options: ‘Not at all,’ ‘several days,’ ‘more than half of the days,’ and ‘nearly every day.’ The total score ranges from 0 to 12, with categories of psychological distress (the term used by PHQ-4 to cover anxiety and depression) being none (0–2), mild (3–5), moderate (6–8), and severe (9–12).

#### 2.4.2. Demographic Variables (Study 1 and 2)

Three sociodemographic variables were measured: (1) Age was assessed continuously in years, (2) gender was assessed using Office for National Statistics binary categories (male, female) [39], (3) shielding category was assessed with the follow item: ‘At the beginning of the pandemic, did you receive a letter or text from your GP or the NHS telling you that you were at risk?’ There were 2 responses: ‘Yes’ or ‘no.’ We did not measure if people did shield, just whether or not they had received a letter or text from government indicating that they should be shielding because they have a medical condition that puts them at risk of severe COVID-19.

#### 2.4.3. Illness Beliefs (Study 1 and 2)

Two illness belief constructs were measured. (1) Threat perception was measured using 2 items to assess the constructs ‘perceived severity’ and ‘perceived vulnerability’: ‘If you were ill with Covid-19 it would be serious for you;’ and ‘It is likely that you will get Covid-19’. There were four responses, ranging from ‘strongly agree’ to ‘strongly disagree’, with the option of responding ‘don’t know’ or ‘prefer not to say.’ In line with the Protection Motivation Theory [40], we multiplied the measures of perceived severity (scale 1–4) and perceived vulnerability (scale 1–4) to produce a total threat perception score (range 1–16). (2) Illness (COVID-19) representations were measured using an adapted brief illness perception questionnaire [41]. The brief illness perception questionnaire uses a single statement to assess constructs from CS-SRM, indicating level of agreement with each statement using a 4-point Likert rating scale. Beliefs about the illness COVID-19 was measured using 5 items: (1) ‘Covid-19 would have major consequences for my life’ (consequences), (2) ‘Covid-19 symptoms would last a long time’ (duration/timeline), (3) ‘You could get Covid-19 again’ (recurrence/timeline), (4) ‘You would spend time worrying about having Covid-19’ (emotional representation), (5) ‘Having Covid-19 would make you feel anxious’ (emotional representation). We calculated the average of these 5 items as a total score for illness representations, with higher scores reflecting more negative representations of COVID-19, Cronbach’s alpha = 0.81 [6].

#### 2.4.4. Environment—Study 1

Three environment variables were measured. (1) Rurality was assessed using an 8-fold urban/rural classification [42]. Participant postcodes were used to classify participants as living in a rural or urban area. The classification is based on size of settlement and drive time to settlements of 10,000 people or more and is as follows: 1 = large urban areas, 2 = other urban areas, 3 = accessible small towns, 4 = remote small towns, 5 = very remote small towns, 6 = accessible rural areas, 7 = remote rural areas, 8 = very remote rural areas. (2) Area deprivation was assessed using the Scottish Index of Multiple Deprivation (SIMD), which looks at the extent to which an area is deprived across seven domains: Income, employment, education, health, access to services, crime, and housing [43]. All 6976 data zones (postcodes) were grouped into 10 bands (deciles), each containing 10% of the data zones. Decile 1 contained the 10% most deprived data zones in Scotland, and decile 10 contained the 10% least deprived data zones in Scotland. (3) Access to residential outside space was assessed with 1 item: ‘Thinking about any outside space you may have at the property you are currently living in (by outside space, I mean a garden, balcony, patio, etc.) do you have access to an outside space at your property?’ There were 4 response options which were recoded to assess outside space: The 2 responses, ‘yes a private outside space only’ or ‘yes a private and shared outside space,’ were recoded 1, ‘yes, a shared outside space only’ was recoded 2, and ‘No’ was re-coded 3. The higher the score, the less access to outside space.

#### 2.4.5. Environment—Study 2

Two variables about green space were measured. (1) Frequency of visiting green space was measured with 1 item: ‘On how many days in the last week did you visit any public green or open spaces, for example a park, countryside, wood, play area, canal path, riverside or beach (private or shared garden not included)?’, with answer options between 0 and 7 days of the week. (2) Duration of visiting green space was measured with 1 item: ‘Thinking about the days in the last week you visited public green or open spaces, how much time, on a typical weekday visit, did you spend in these public green or open spaces?’, with 5 answer options: ‘Up to 10 min,’ ‘11 up to 30 min,’ ‘30 min up to 1 h,’ ‘1 up to 2 h,’ and ‘2 h or more.’ The study also included a potential covariate, namely distance to green space, which was measured with 1 item: ‘How far away from your home is your nearest public green or open space?’ There were 6 response options: ‘Less than a 5 min walk,’ ‘within a 5–10 min walk,’ ‘within an 11–20 min walk,’ ‘within a 21–30 min walk,’ ‘more than a 30 min walk,’ and ‘none within walking distance.’ These items were based on access to green space indicators in the Scottish Household Survey [37].

### 2.5. Data Collection

Ipsos MORI administered the self-reported questionnaire by conducting telephone interviews using Computer Aided Telephone Interviewing (CATI).

### 2.6. Statistical Methods

The data were analyzed using SPSS version 25.0. For all variables, the answers ‘I don’t know’ and ‘I prefer not to say’ were treated as missing values and, therefore, excluded from the analyses. Most data only had few missing values, which were managed by listwise deletion of cases in any given analyses. *p*-values of *p* < 0.05 were taken as statistically significant. To test whether the effects of the individual demographic and illness beliefs variables on mental health were moderated by environment variables, moderation models with Hayes’ PROCESS macro (v 3.5, model 1) [44] were used. The SPSS macro PROCESS runs each predictor construct individually. The moderation models were tested in 2 steps. In the first step, 2 variables were included, and the effects of demographic independent variables or illness beliefs variables were tested with an environment moderator variable. In the second step, the interaction term between the independent variable and the moderator variable was entered. For the analyses, a 95% bias-corrected percentile bootstrapped confidence interval (CI) method was used, and 5,000 bootstrap resamples were produced for moderation examination. We employed conventional methods for plotting simple slopes to understand moderation effects at 1 standard deviation below and above the mean [45].

## 3. Results

### 3.1. Study 1

#### 3.1.1. Participants

Table 1 shows that there were 2969 participants in Study 1, of which 1765 (59.6%) were female, 349 (11.9%) had received a letter or text stating that they should be shielding, and the median age was 54 years. Overall, this brings all percentages within the quotas, accounting for 30% leeway. Table 1 shows that 1757 (67%) lived in urban areas (1 and 2 classification), 350 (13.3%) lived in small towns (3–5 classification), and 515 (19.6%) lived in rural areas (6–8 classification). In the current study, 133 participants (5.1%) lived in the most deprived areas of Scotland, and 352 (13.4%) lived in the least deprived areas. The overwhelming majority (91.6%) had access to private outside space. Most participants (*n* = 2650; 89.3%) did not report psychological distress.

#### 3.1.2. Direct Effects of Individual Factors on Psychological Distress

There were significant linear associations between age and psychological distress, with young people more psychologically distressed; gender and psychological distress, with females more psychologically distressed; and shielding category and psychological distress, with people who had received a letter or text stating that they should be shielding being more psychologically distressed (Table 2). There were significant associations between illness (COVID-19) representations and psychological distress and between threat perception and psychological distress, with worse beliefs about the illness COVID-19 and greater threat perception being associated with more psychological distress.

#### 3.1.3. Direct Effects of Environmental Factors on Psychological Distress

There were significant linear associations between rurality and psychological distress, with people living in more rural areas less psychologically distressed; area deprivation and psychological distress, with people from the least deprived areas less psychologically distressed; and access to residential outside space and psychological distress, with people being more psychologically distressed with access to shared outside space or no outside space compared to people with private outside space (Table 2).

#### 3.1.4. Moderator Effects of Environment on Demographic Variables and Psychological Distress

Rurality did not moderate the relationship between any of the three demographic variables (age, gender, shielding category) and psychological distress (Table 3 and Figure 1). As described above in the Methods section, we used the eight-fold rural/urban classification. We also tested the moderator effect with the six-, three-, and two-fold rural/urban classification [42], which also showed no moderator effects.

Area deprivation moderated some relationships between demographic independent variables and the dependent variable, psychological distress. Area deprivation moderated the relationship between age and psychological distress. A follow-up simple slope analysis demonstrated that the negative association between age and psychological distress (older people experiencing less psychological distress) was more pronounced in people from the least deprived areas (Table 3). The association between gender and psychological distress was also moderated by area deprivation. The follow-up simple slope analysis demonstrated that males experiencing less psychological distress than females was more pronounced in people from the least deprived areas. Area deprivation did not moderate the effect of being in the group that had received a letter or text stating that they should be shielding and psychological distress.

Access to residential outside space moderated some relationships between demographic independent variables and the dependent variable, psychological distress. The association between age and psychological distress was not moderated by access to outside space, nor was the association between gender and psychological distress. The association between shielding category and psychological distress was moderated by access to shared outside space (Table 3). The follow-up simple slope analysis demonstrated that the effect of receiving a letter or text stating informing the participant that they were in the shielding category on psychological distress as follows: Psychological distress was higher for people who received the letter/text, and this effect was more pronounced for people who had shared outside space compared to people who had access to private outside space. Further, people who had received a letter/text did not have more psychological distress than people who had not received a letter/text among the group who did not have any outside space (see Table 3 and Figure 1).

#### 3.1.5. Moderator Effects of Environment on Illness Beliefs and Psychological Distress

Rurality did not moderate the relationship between illness (COVID-19) representations or threat perception and psychological distress.

Area deprivation moderated the association between illness (COVID-19) representations and psychological distress. The moderation explained a significant increase in variance in psychological distress. A follow-up simple slope analysis demonstrated that the positive association between illness (COVID-19) representations and psychological distress was more pronounced in more deprived areas than more affluent areas (Table 2). The association between perceived threat and psychological distress was not moderated by area deprivation.

Access to residential outside space did not moderate the association between illness (COVID-19) representations and psychological distress but did moderate the association between threat perception and psychological distress (Table 2). The follow-up simple slope analysis demonstrated that the positive association between threat perception and psychological distress was more pronounced for people with no outside space, followed by people with shared space only and people with private space.

### 3.2. Study 2

#### 3.2.1. Participants

Table 1 shows that there were 502 participants in Study 2, of which 295 (58.60%) were female, 58 (11.6%) had received a letter or text stating that they were in the shielding category, and the median age was 53 years. Accounting for the 30% leeway, the quotas were met. Just under one-half of participants either did not visit green space or visited only once during a week, and just over one-fifth visited green space every day of a typical week. Over one-half were less than 5 min away from home to green space. The most common duration of time spent in green space was 30 min to 60 min, with 39.4% of participants stating this duration. Most participants (*n* = 444; 88.8%) did not report psychological distress.

#### 3.2.2. Direct Effects of Individual Factors and on Psychological Distress

As per Study 1, there were significant linear associations between age and psychological distress, with young people more psychologically distressed; gender and psychological distress, with females more psychologically distressed; and receiving a letter or text informing the participant that they were in the shielding category and psychological distress, with people who had received a letter/text about shielding being more psychologically distressed. There were also significant associations between illness (COVID-19) representations and psychological distress and between threat perception and psychological distress, with worse beliefs about the illness COVID-19 and greater threat perception being associated with more psychological distress (Table 2).

#### 3.2.3. Direct Effects of Green Space on Psychological Distress

There was a significant linear association between frequency of visiting green space and psychological distress; visiting green space more often was associated with less psychological distress. The association between duration of visiting green space and psychological distress was nonsignificant (Table 2).

#### 3.2.4. Moderator Effects of Green Space on Demographic Variables and Psychological Distress

Frequency of visiting green space did not moderate the effects of age and gender but did moderate the effects of shielding category (Table 2 and Figure 1). The follow-up simple slope analysis demonstrated that the association between receiving a letter or text informing the participant that they were in the shielding category and greater psychological distress was more pronounced in people who more frequently visited green space. In the group of people who visited green space less frequently (1 standard deviation (SD) below the mean), people who had received a letter/text about shielding no longer differed in psychological distress from those who had not received a letter/text about shielding.

Duration of visiting green space did not moderate the effects of age and gender but did moderate the effect of shielding category (Table 2 and Figure 1). The follow-up simple slope analysis demonstrated that the association between receiving a letter/text about shielding and greater psychological distress was more pronounced in people who visited green space for shorter durations. In the group of people who visited green space for longer durations (1 SD above the mean), people who had received a letter/text about shielding no longer differed in psychological distress from those who had not received a letter/text about shielding.

#### 3.2.5. Moderator Effects of Green Space on Illness Beliefs and Psychological Distress

Neither frequency nor duration of visiting green space moderated the association between both illness (COVID-19) representations and threat perception and psychological distress.

## 4. Discussion

### 4.1. Key Results

Whereas the interactions between individual and environmental variables were statistically significant, they explained only a little of the variance in psychological distress. Nonetheless, in each case of statistically significant moderation, being in an area of deprivation, lacking private residential outside space, and being in green space worsened the direct effects of the individual factors. Thus, these environmental factors were not only directly associated with poorer mental health but also appeared to exacerbate the effects of detrimental individual factors. These findings therefore suggest that environment is a significant factor that has influenced mental health during the COVID-19 crisis. However, this interpretation is limited by the cross-sectional nature of the data. For example, it is also possible that people with better mental health made more use of green space.

### 4.2. Consistency with Other Studies and Implications

The study found that people living in urban areas had worse psychological distress compared to people living in rural areas, thereby supporting previous pre-pandemic research summarised in systematic reviews [19,20] and a study of rural and urban differences in China during the COVID-19 pandemic [46]. However, we found no rural/urban moderation effects. Mental health covers a range of definitions and mental disorders and it may be that psychological distress, which was measured in this study, does not significantly differ between rural and urban areas when other factors are considered, or at least does not in Scotland during the current pandemic at the time when data were collected. Nevertheless, research in Scotland suggests that if there are urban and rural differences in mental health then it is likely to vary by type of mental disorder. For example, one pre-pandemic study found no differences in schizophrenia by degree of urbanisation [47] but other studies found higher rates of death by suicide in rural compared to urban areas [48,49]. Further research focusing on rural and urban differences in mental health during the COVID-19 pandemic should therefore consider focusing on relevant and specific mental disorders, i.e., where rural and urban differences are already known and are likely to be negatively impacted by the COVID-19 crisis. In Scotland, for instance, monitoring rural and urban differences and the impact of the COVID-19 crisis on death by suicide might be relevant. This may be particularly important if the economic impact of the COVID-19 crisis results in unemployment and hardship because there is a concern that the number of deaths by suicide will increase, which could impact rural communities who already have higher incidence of death by suicide, as well as deprived areas that are likely to bear the brunt of an economic recession [50,51].

We found differences in psychological distress in deprived compared to affluent areas and show that females and people with worse illness (COVID-19) representations who live in deprived areas may have worse psychological distress during the COVID-19 crisis. Given that there is strong evidence summarised in systematic reviews that some types of mental health are worse in deprived areas [15,16,17] and other studies suggesting that mental health during the COVID-19 crisis is worse in deprived areas [5,6] this is not surprising.

Our study suggests that one potential longer-term environmental strategy that could be used to improve mental health during pandemics is improving access to residential outside space. The study, for example, found that mental health was worse for people who shared outside space who had received a letter/text informing them that they were in the shielding category (incidentally a group at greater risk of poor mental health due to being in quarantine [52]) compared to those who shared an outside space who were not in the shielding category. The results showed that the association between threat perception and mental health was worse for people who did not have access to any residential outside space. UK studies have suggested regional variations in access to residential outside space and use of green space during the COVID-19 crisis [53,54]. In London, 74% of people have access to a garden compared to 97% in Wales, closely followed by 96% of people in the East Midlands [53]. Households with single adults living alone had the least access to gardens (79%) during lockdown compared to households with children or with two or more adults (91%) [53]. These differences in access to residential outdoor space may prove significant for mental health. The importance of the immediate home environment was found in a recent survey of 8177 students from a university institute in an Italian region most heavily hit by COVID-19.The survey found that living in small apartments with poor views was associated with an increased risk of moderate-severe and severe depressive symptoms [55]. The importance of good-quality housing as a contributor to good mental health was found in a systematic review that summarized evidence prior to the pandemic of the relationship between building and health [56]. During the current pandemic, the degree of access people have to outdoor space in the immediate home environment, and whether it is private or shared is therefore important. Given the evidence that living in urban versus rural areas and living in areas of high deprivation are associated with worse psychological distress, those living in homes without outdoor spaces in deprived urban areas may be particularly vulnerable, especially as these locations are also less likely to have public green space within a 5-minute walk [36]. It is difficult in the short term to alter the kind of residential outdoor space that people have access to in existing housing. Therefore, longer-term strategies are needed to effectively address such issues. However, some short-term strategies could be effective during a pandemic, such as offering private access to shared residential space by operating a timed rotation for different households. In some situations, it may also be possible to make adjacent areas of publicly owned outdoor space temporarily available to single households only.

Another potential environment strategy that could be harnessed to improve mental health during the current COVID-19 crisis is improving access to public green space. Research summarized in systematic reviews suggests that adults who live in neighborhoods with a greater quantity of green space around the residential environment report better mental health than adults who live in less green neighborhoods [57]. Our study produces a complex picture in relation to public green space, with greater frequency of visits supporting better mental health in general. However, frequently visiting green space exacerbated the effect of being in the shielding category and worse psychological distress, whereas there were no differences in psychological distress between people who were in and not in the shielding category among those who did not frequently visit green space. One possible explanation is that people in the shielding category who were frequent visitors to green space may experience greater anxiety in public spaces compared to people not in the shielding category because of concerns about being able to keep a 2-meter distance from other people. Of course, if one does not frequently visit green space, then this problem of keeping a 2-meter distance is no longer relevant. Spending only a short duration in green space also exacerbated the effect of being in the shielding category and worse psychological distress, whereas there were no differences in psychological distress between people who were in and not in the shielding category among those who spent a longer in green space. This finding may reflect anxiety associated with being in a public space for people in the shielding category, who might only make short visits because of fear of not being able to adequately distance from other members of the public. Closure of some parks and restrictions on movement might have exacerbated potential problems of crowding and contributed to poorer mental health outcomes during the pandemic [58], particularly for those with no residential outside space. A recent survey found that half of people in London visited a park or public green space in the past 7 days during April 2020, which was well above the average of 34% across Britain [53]. However, data from a UK-wide survey undertaken under lockdown at the end of April 2020 and in Scotland from 29 May–5 June 2020 [59] showed similar patterns: More people reduced the time spent outdoors or in green space than increased time there while under lockdown [60]. The most marked reduction in time spent in green and open spaces was among older people, who may be likely to perceive themselves most vulnerable to COVID-19. Nonetheless, the majority of participants (63–65%) reported that green and open space benefitted their mental health during the lockdown period, and this was slightly more likely for females and young adults (25 to 44 years old) and those of a higher socioeconomic grade. Our study was undertaken later on, in early July, when Scotland was undergoing a phased lifting of restrictions on movement. Although 84% of our sample had public green space within a 5–10-minute walk, nearly 36% had not visited green space in the previous week. Given the association with reduced psychological distress for frequency of green space visits, maintaining public open space access during the pandemic is clearly important. Moreover, facilitating its safe use by those in the shielding category, e.g., by having well-managed physical distancing measures enforced, could be an important contribution to mental wellbeing.

### 4.3. Limitations

Current findings should be interpreted in light of key limitations. The study shows that environment exaggerated the effect of some demographics and illness beliefs on mental health, thereby offering some explanation for the widening health inequalities apparent during COVID-19 and possible opportunities for ameliorating these effects during the current and any future pandemics. Nonetheless, we did not include the full breadth of demographic factors, for example, employment status and social-economic status. The environment only explained a small amount of the variance in mental health, and much of the variance in mental health remains unexplained. Second, the current study used only a brief timeframe to measure associations between environment and mental health. Whether environment can attenuate the effects of the COVID-19 pandemic on mental health in the longer term cannot be deduced and merits further investigation. Third, the use of a cross-sectional design precludes the inference of causal effects among the variables. Inferences about relationships between the variables are therefore premised on theoretical suppositions and previous empirical evidence and cannot be inferred from the actual study data. Hence, prospective, longitudinal studies and, where feasible, randomized experiments, are needed to examine the potential causal effects of environment on mental health during pandemics. Related to this point, we did not examine mechanisms to explain environmental differences and mental health. Future research should consider measuring proposed potential mechanisms such as the effect of environment on mood and recovery from stress [61] and environment effects on social support and interpersonal relationships [62,63]. Both of these potential mechanisms may be pertinent in pandemics due to the effects of communicable disease on stress and on restrictions on social interactions to reduce transmission of the infection. Fourth, while objectives measures of rurality and area deprivation were used, we measured perceptions of distance to green space rather than objective measurement. There is (Scottish) evidence that there is limited agreement between perceived and objective measures of distance to green space. How people perceived distance to green space during the pandemic seemed particularly relevant to understanding the effects of access to green space on mental health. Nonetheless, future studies should consider the utility of objective measurements of access to green space [64].

## 5. Conclusions

Environment is important for mental health. This has implications both for public health and for environmental planning, design, and management, including housing design and public open space provision and regulation. Opportunities exist for supporting mental health by encouraging the use of existing residential outside space and public green space and facilitating safe use, even for those in the shielding category. Future research about mental health during the current COVID-19 crisis should include the environment as a key concept in conceptual models. A multidisciplinary approach should be used to inform public mental health strategies and research to bring psychological and environmental insights about mental health during pandemics.

## Figures and Tables

**Figure 1 ijerph-18-03869-f001:**
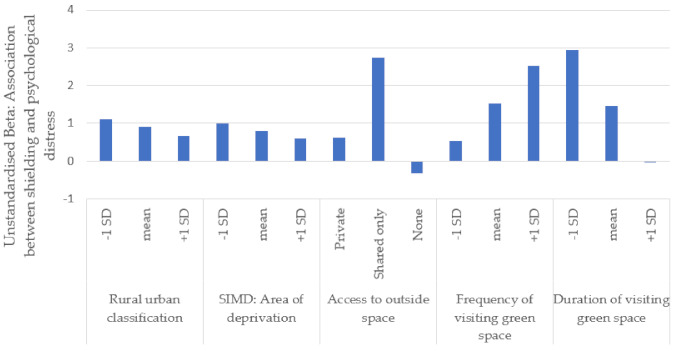
Effects of moderations between shielding category and the environment (Study 1 and Study 2) on psychological distress.

**Table 1 ijerph-18-03869-t001:** Characteristics of people who participated in Study 1 and 2.

		Study 1		Study 2	
		*n*	%	*n*	%
Total *n*		2969	100	502	100

Demographic variables					
Age (in years)	Median, IQR	54	38–65	53	38–65
Gender	Male	1198	40.4	206	41.1
	Female	1765	59.6	295	58.9
In shielding category	Yes	349	11.9	58	11.6
	No	2592	88.1	442	88.4

Illness beliefs					
Threat perception (range 1–16)	Median, IQR	6.0	4.0–8.0	6.0	4.0–8.0
Illness (COVID-19) representation	Median, IQR	3.2	2.8–3.6	3.2	2.6–3.6

Environmental factors					
Rural Urban classification	1 (large urban area)	901	34.4		
	2	856	32.6		
	3	218	8.3		
	4	84	3.2		
	5	48	1.8		
	6	337	12.9		
	7	93	3.5		
	8 (very remote rural area)	85	3.2		

SIMD	1 (10% most deprived)	133	5.1		
	2	176	6.7		
	3	198	7.6		
	4	216	8.2		
	5	244	9.3		
	6	297	11.3		
	7	317	12.1		
	8	348	13.3		
	9	341	13.0		
	10 (10% least deprived)	352	13.4		

Access to residential outside space	Yes, private	2254	91.6		
	Yes, shared only	157	6.4		
	No	51	2.1		

Frequency visiting green space	0 (days per week)			180	35.9
	1			60	12.0
	2			51	10.2
	3			30	6.0
	4			29	5.8
	5			21	4.2
	6			18	3.6
	7 (days per week)			113	22.5

Distance to green space from home	Less than a 5 min walk			284	56.7
	Within a 5–10 min walk			139	27.7
	Within an 11–20 min walk			52	10.4
	Within a 21–30 min walk			10	2.0
	More than a 30 min walk			7	1.4
	None within walking distance			9	1.8

Duration visiting green space	Up to 10 min			22	6.8
	11 up to 30 min			49	15.2
	30 min up to 1 h			127	39.4
	1 up to 2 h			86	26.7
	2 h or more			38	11.8

Outcome variable					
Psychological distress	Median, IQR	1.0	0.0–3.0	1.0	0.0–3.0
	Normal: Lower than 6	2650	89.3	444	88.8
	Probable case: 6 or higher	316	10.7	56	11.2

Note: Total numbers do not always add up to 2969 and 502 due to missing data. IQR: Interquartile range. SIMD: the Scottish Index of Multiple Deprivation.

**Table 2 ijerph-18-03869-t002:** Linear regression effects of individual demographics and illness beliefs and environmental factors on psychological distress.

			Standardised Beta	R^2^	*p*
Study 1	Individual demographics and illness beliefs	Age	0.16	0.024	<0.001
		Gender [Male dummy coded]	0.14	0.020	<0.001
		In shielding category by the government [Shielded dummy coded]	0.12	0.014	<0.001
		Illness (COVID-19) representation	0.17	0.030	<0.001
		Threat perception	0.13	0.017	<0.001
	Environmental factors	Rurality ^a^	0.04	0.002	<0.05
		Area deprivation ^b^	0.13	0.016	<0.001
		Access to residential outside space		0.016	<0.001
		Shared outside space	0.11		<0.001
		No outside space	0.07		<0.01
Study 2	Individual demographics and illness beliefs	Age	0.20	0.039	<0.001
		Gender [Male dummy coded]	0.22	0.047	<0.001
		In shielding category by the government [Shielded dummy coded]	0.17	0.028	<0.001
		Illness (COVID-19) representation	0.20	0.041	<0.001
		Threat perception	0.16	0.025	<0.001
	Environmental factors	Frequency visiting green space	0.18	0.031	<0.001
		Duration visiting green space	0.11	0.012	<0.06

^a^ Eight-fold rural/urban classification. ^b^ Scottish Index of Multiple Deprivation.

**Table 3 ijerph-18-03869-t003:** Moderation of individual demographics and illness beliefs by environmental factors.

		Individual Factors														
		Age			Gender			In Shielding Category by the Government			Illness (COVID-19) Representation			Threat Perception		
Study 1: Environmental factors		ΔR2	F, *p*	simple slope −1sd, 0, +1sd	ΔR2	F, *p*	simple slope −1sd, 0, +1sd	ΔR2	F, *p*	simple slope −1sd, 0, +1sd	ΔR2	F, *p*	simple slope −1sd, 0, +1sd	ΔR2	F, *p*	simple slope −1sd, 0, +1sd
	Rurality ^a^		Ns.			Ns.			Ns.			Ns.			Ns.	
	Area deprivation ^b^	0.0014	3.93, <0.05	0.02 0.02 0.03	0.0016	4.40, <0.05	0.96 0.74 0.51		Ns.		0.0094	25.21, <0.001	1.03 0.64 0.26		Ns.	
	Access to residential outside space ^c^		Ns.			Ns.		0.0075	9.33, <0.001	0.61 2.74 0.32		Ns.		0.0031	3.12, <0.05	0.10 0.23 0.32
Study 2: Environmental factors	Frequency visiting green space		Ns.			Ns.		0.0128	6.78, <0.01	0.53 1.52 2.53		Ns.			Ns.	
	Duration visiting green space		Ns.			Ns.		0.0263	9.07, <0.01	3.11 1.52 0.08		Ns.			Ns.	

^a^ Eight-fold rural/urban classification. ^b^ Scottish Index of Multiple Deprivation. ^c^ Reported simple slopes reflect having access to private outside space, shared outside space only, no outside space.

## Data Availability

The data that support the findings of this study are available on request from the corresponding author. The data are not currently publicly available due to the research team still publishing from these data.

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
