# Peer review of "Are Rurality, Area Deprivation, Access to Outside Space, and Green Space Associated with Mental Health during the COVID-19 Pandemic? A Cross Sectional Study (CHARIS-E)"

_ijerph, 2021, doi:10.3390/ijerph18083869_

Round 1

Reviewer 1 Report

I rarely do a review like this, but in this case I have to write it: it is a very good manuscript! It has a few small flaws that I proposed to the authors for consideration. In this article, the authors decided to investigate if rurality, area deprivation, access to outside space and frequency of visiting and duration in green space are associated with mental health during the Covid-19 pandemic. They divided their research into two parts.

The obtained results are interesting and presented clearly, and the applied statistical methods are correct. In my opinion, the limitations mentioned by the authors are not an obstacle to the publication of these results. I believe that they are a good indication of our knowledge gaps in this field, and I believe that the authors will undertake the research they write about in the future.

Minor comments: 

  1. The title is too long and very extensive.
  2. L48: Is the way to quote in () correct? Shouldn't you use [] ??
  3. eg. L56 I think double spaces appear in random places.
  4. Another strong point of this research is the large sample size.
  5. The discussion is terribly long, it could be shortened. It makes no sense to cover everything so thoroughly. It is possible to lose the main thoughts in it.

Reviewer 2 Report

Thank you for the opportunity to review this manuscript. Overall, I feel it is generally well-written, sound analysis and addresses a timely and important topic--namely COVID-19, mental health and environmental factors. There are a number of strengths of the study; however, I have some relatively minor suggestions to further strengthen it for publication:

  1. The Introduction mentions the CS-SRM (p2, line 61) but fails to provide the reader a more thorough description of this model. I would suggest adding more about the CS-SRM in the Introduction, especially as it is used as the basis for some of the study measures. 
  2. I would also suggest the authors describe what is meant by "Covid-19 shielding status" in the Introduction since it is not a term that may be used in other countries. While there is some information about how this was assessed in the Measures section, I would suggest defining/explaining it when it is first introduced in the Introduction.
  3. Please provide more information about how missing data was addressed in the analyses.
  4. Did authors consider examining or controlling for impact of employment or SES status in the analyses? At minimum, this should be mentioned as a limitation to the study.
  5. Misc. Grammatical Issues:
    1. "area" in title should be capitalized 
    2. Replace "confirm" with "examine" in Abstract, p 1, line23
    3. Consider removing the report of demographic info in the Abstract (p 1, lines 30-32), or making sure is complete sentences--i.e., "(59.6%) female" should be "were female", etc.
    4. Change "environment" to "environmental factors" (p 1, line 36)
    5. "Psycho-evolutionary" and "Biophilia hypothesis" do not need to be capitalized (p 2, line79-80)
    6. change "resident" to "residents", p 3, line 128

Reviewer 3 Report

This paper is just timely and excellent!

Nowadays, all over the world, people have been worried and living in stressful daily lives by COVID-19.

This paper precisely shows the condition and importance of the green space.

It will be a rationale paper to create positive living style and environment against COVID-19!   
